# Prevalence and factors influencing drug-resistant tuberculosis in four regions of Ghana

**Esther Ba-Iredire[1], James Atampiiga Avoka[2]\*, Luke Abanga[3], Abigail Awaitey Darkie[4], Emmanuel Junior Attombo[5], Eric Agboli[6]**

1 Eastern Regional Hospital, Koforidua, Ghana, 2 Lower Manya Krobo Municipal Health Directorate, Odumase Krobo-Eastern Region, Ghana, 3 Luke Abanga, Jirapa Municipal Health Directorate, Upper West Region, Ghana, 4 Abigail Awaitey Darkie, Nkawkaw Health Center, Eastern Region, Ghana, 5 Emmanuel Junior Attombo, Municipal Health Directoratye, Central Region, Ghana, 6 University of Health and Allied Sciences, Volta Region-Ho, Ghana

\* avokajames@yahoo.com, avokaj49@gmail.com

## Abstract

### Introduction

The alarming rate of drug-resistant tuberculosis (DR-TB) globally is a threat to treatment success among positive tuberculosis (TB) cases. Studies aimed at determining the prevalence, trend of DR-TB and socio-demographic and clinical risk factors contributing to DR-TB in the four regions of Ghana are currently unknown. This study sought to determine the prevalence and trend of DR-TB, identify socio-demographic and clinical risk factors that influence DR-TB, and analyse the relationship between underweight and adverse drug reactions and treatment outcomes among DR-TB patients in four regions of Ghana.

### Method

It was a retrospective review conducted over 5 years, from January 2018 to the end of December 2022. The data were retrieved from the DR-TB registers and folders at the Directly Observed Treatment (DOT) centres in the four regions. Analysis of the data was conducted using STATA version 17.

### Results

The prevalence of DR-TB in Ashanti was 10.1%, Eastern 5.3%, 27.8% in Central, and 2.7% in the Upper West region for the year 2022. The overall prevalence rate of DR-TB for the period 2018–2022 was 13.8%. The socio-demographic and clinical risk factors that influence DR-TB in the four regions are: age, marital status (aOR 3.58, P-value< 0.00, 95% CI 2.86–4.48), Senior High School (SHS) level of education (aOR 2.09, P-value=0.01, 95% CI 1.21–3.63), alcohol intake (aOR 0.49, P-value <0.00, 95% CI 0.38–0.63), previously treated (aOR 22.03, P-value<0.00,

**Data availability statement:** All relevant data are within the manuscript and its Supporting Information files.

**Funding:** The author(s) received no specific funding for this work.

**Competing interests:** The authors have declared that no competing interests exist.

**Abbreviations:** aOR, Adjusted Odds Ratio; BMI, Body Mass Index; CDC, Centre for Disease Control; CI, Confidence Interval; COR, Crude Odds Ratio; DOT, Directly Observed Treatment; DR, TB, Drug Resistant Tuberculosis; DST, Drug Sensitivity Test; ECDC, European Center for Disease Prevention and Control; HIV, Human Immuno, deficiency Virus; MDR, TB, Multi Drug Resistant Tuberculosis; NTP, National Tuberculosis Program; OR, Odds Ratio; PAHO, Pan, American Health Organization; RR, TB, Rifampicin Resistant Tuberculosis; SHS, Senior High School; TB, Tuberculosis; UHAS, University of Health and Allied Sciences; UI, Uncertainty interval; WHO, World Health Organization; XDR, TB, Extremely Drug Resistant Tuberculosis.

CI 16.58–29.26), major adverse drug reaction (aOR 125.50, P-value<0.00, 95% CI 58.05–271.34), and minor adverse drug reaction (aOR 23.59, P-value<0.00, 95% CI 18.32–30.39); treatment outcome, cure (aOR 0.52, P-value<0.00, 95% CI 0.41–0.66), completed (aOR 9.67, P-value<0.00, 95% CI 6.56–14.28), relapsed (aOR 2.62, P-value=0.01, 95% CI 1.33–5.18), Lost-to-Follow-up (LTFU) (aOR 0.45, P-value<0.00, 95% CI 0.29–0.70), and failure (aOR 35.24, P-value<0.00, 95% CI 7.76–159.99). Also, there was an association between underweight and adverse drug reaction (RRR 5.74, P-value<0.00, 95% CI 4.86–6.79) and treatment outcome (RRR 0.79, P-value<0.00, 95% CI 0.74–0.86).

## Conclusion

The study shows that the prevalence of DR-TB in Ghana is low, probably not because the cases have reduced but due to inadequate GeneXpert machines to detect the cases. Age, marital status, education, alcohol intake, previously treated TB cases, adverse drug reactions, underweight, and treatment outcome are factors influencing the development of DR-TB. Therefore, interventions aimed at improving the nutritional status of DR-TB cases and minimising adverse drug reactions will improve treatment outcomes.

---

## Background

The End TB strategy, which was adopted by the WHO in 2014, aims to reduce 90% of all TB deaths and the incidence rate by the end of 2030 [1]. Significant among the Sustainable Development Goals (SDGs) is the target of ending the unbearable worldwide TB disease [2]. According to the WHO, 2017 was a defining moment to combat DR-TB. The DR-TB emergence poses a huge burden on TB management in Africa and the world at large [3] and therefore requires extra effort in research and innovation to achieve the set targets. Since 1994, the World Health Organisation (WHO) has systematically collected and analysed data on levels of resistance to anti-TB drugs from countries and territories. The unfortunate development of drug-resistant tuberculosis (DR-TB) has become a threat to TB treatment success rates the world over [4]. Approximately 214,000 deaths were recorded in 2018 as a result of MDR/rifampicin-resistant TB (RR-TB), with another 484,000 incident cases comprising 3.4% of the incident cases and 18% of the ever-treated TB cases recorded globally [5]. Studies have shown that TB treatment failure and poor management outcomes are rampant among previously treated MDR-TB cases [5–7]. Treatment failure prolongs the treatment time, and this complicates the management of the case, which is dangerous to the clients [8,9]. Currently, DR-TB cases are detected using the Gene-Expert machine as first line and culture and sensitivity test (DST) for the confirmation. Sputum samples are collected and transported to the laboratory for GeneExpert analysis. The Genexpert test is a molecular test that detects the DNA in TB bacteria. It uses a sputum sample and can give results within 2 hours. It can also detect the

genetic mutations associated with resistance to the drug Rifampicin [10]. A study shows that the African region recorded the highest cases of DR-TB which has significantly contributed to the TB deaths [11]. A study in Ghana shows that 14.4% of TB cases are said to have developed resistance to rifampicin [12]. Against this backdrop, this study sought to investigate the prevalence and factors that influence drug-resistant TB in four (4) regions of Ghana towards charting new paths and opportunities for achieving higher treatment success rates to halt the spread of the disease globally.

## Materials and methods

### Study design

This is a retrospective review that was conducted from 2018 to 2022. The study was conducted among DR-TB cases enrolled for treatment in all the treatment centres in the Ashanti, Central, Eastern, and Upper West Regions of Ghana. These cases were registered and reported for treatment at the various treatment sites in the four (4) study regions. All the cases were on supervised treatment with second-line drugs in addition to the periodic provision of enablers in the form of food and transport motivations. Treatment progress was closely supervised and monitored monthly using microscopy, smears, and sputum culture analysis. A DR-TB case is defined as a detected TB case that develops resistance to any of the first-line anti-TB drugs (rifampicin, isoniazid, or any other of the first-line TB drugs). After the extraction, a logistic regression model was used to determine the association between the predictor variables and the outcomes at 5% confidence intervals.

### Study variables

The outcome variables in this study are prevalence of DR-TB and DR-TB cases that are underweight. The independent variables are age, sex, marital status, education level, religion, Body Mass Index (underweight), previously treated TB cases, chronic conditions, smoking, alcohol intake, adverse drug reaction (major and minor), and treatment outcome. In this study, major adverse drug reactions are drug reactions that require hospitalization, are life-threatening, results in death or incapacitation. Minor adverse drug reactions do not require hospitalization and are not life-threatening but any untoward event attributable to the drugs.

### Data extraction

Data extraction sheets were developed to extract the information on all cases, taking into account their demographic data such as age, sex, educational background, ethnicity, employment status, and marital status. The data were extracted from the medical records such as DR-TB registers, treatment folders, and the District Health Information Management System 2 (DHIMS2) of the patients daily from Mondays to Fridays for two consecutive weeks, from June 5–16, 2023. The authors did not have access to information that could identify individual participants during or after the data collection. Additionally, other variables such as smoking, alcohol intake, previous treatment, Body Mass Index (BMI), and any chronic conditions were extracted, including treatment outcomes. The main outcome variable was the prevalence of DR-TB in the four study sites. Additionally, underweight DR-TB cases were also used as outcome variable. A weighing scale and a height measure were used to calculate the weight and the height in metres. This was accomplished by calculating the DR-TB cases weight in kilogrammes and height in meters. The BMI was then determined using the formula: BMI = weight (kg)/height (m2). The BMI values were contrasted with the WHO classification guidelines.

### Ethics approval

Ethical clearance was obtained from the Ethics Review Board of the University of Health and Allied Sciences (UHAS) institute for health research, UHAS-REC B.10 [040]22-23. Approval was also obtained from the various treatment sites in the regions where data were obtained, before the study commenced.

## Sampling

The study sites were divided into four zones: The Northern, Southern, Coastal, and Middle Belts. The Bono, Bono East, Ashanti, and Ahafo regions made up the middle belt. The Western, Western North and Central Regions were part of the coastal belt. The Northern, Savanna, Upper East, Upper West and North East Regions were all part of the Northern Belt. The Eastern, Volta, Oti, and Greater Accra regions made up the southern belt. Following that, a random selection of treatment centers was made from each of the four clusters to include them in the study. As a result, Ashanti region was randomly selected to represent the Middle belt, Central Region was selected to represent the Coastal belt, Volta Region represented the Southern belt, and Upper West Region represented the Northern belt. Additionally, in each of the selected regions, the central treatment centers were purposively selected for the study due to the case load. This is because all the DR-TB cases report to those treatment centers for management.

## Study period

The study extracted data on all confirmed TB cases in the four (4) study sites in Ghana spanning from 2018 to December, 2022 with special emphasis on DR-TB.

## Study population

The study population was all confirmed TB cases (high-risk group) in the four study sites. Therefore, the prevalence in this case was calculated as all DR-TB cases identified in the study sites divided by the total number of high-risk population (confirmed TB cases) in the study areas.

$$\text{Prevalence} = \frac{\textbf{DR} - \textbf{TB cases}}{\textbf{Diagnosed TB cases}}$$

## Inclusion and exclusion criteria

The inclusion criteria was all detected DR-TB cases in the four (4) study regions of Ghana who reported at the treatment centres for management and follow-up visits. All DR-TB cases out of all TB cases were registered for the study. Also, records of DR-TB cases before 2017 and after 2023 were excluded. Additionally, cases that were newly diagnosed and not on treatment were not part of this study.

## DHIMS2 data platform

The information is made up of TB cases that report each month and entered into the District Health Information Management System 2 (DHIMS2). In order to assist health managers and decision-makers in collecting and appropriately analysing data at all levels of the health system, the Ghana Health Service developed and expanded the DHIMS2 in 2008 [13,14]. The goal included planning, ssmaking decisions, and allocating resources for the health sector. It was put into effect nationwide in every region and district. The DHIMS2 was created for the storage and reporting of health data. All national health care indicators' data are kept in the DHIMS2 system, from which part of the data were extracted.

## Definition of key terminologies

**Cured:** After 12 months of treatment, the patient must have at least three negative cultures in order to be considered cured.

   **Treatment completion:** Patient who received the prescribed amount of treatment in the allotted time, with positive clinical and radiographic development, but without the necessity of follow-up.

**Failure:** Two or more recommended positive cultures out of three, or three positive cultures in a row, at least 30 days apart, after the 12th month of treatment. The choice to change treatment early due to clinical and radiological deterioration may also take this into account.

**Default:** When the medication was stopped for at least 30 months without interruption by default.

**Death:** When a patient passed away while receiving treatment, for any reason.

**Unfavourable outcome:** The total number of patients whose outcomes were listed as failure, default, or death.

**Treatment success** is the total number of patients whose conditions were determined to be treated successfully and were deemed cured.

**Chronic condition:** For the purpose of this study, chronic condition was defined as hypertension, diabetes, HIV and cardiovascular diseases.

### Data analysis

Analysis of the data was conducted using STATA version 17. Data were entered into Excel sheets, cleaned, and imported into STATA before being analysed. Descriptive analysis was conducted to show the frequencies and percentages of the demographic and clinical risk factors in a tabular form. Thereafter, bivariate analysis was conducted to determine the association between age, sex, marital status, education level, smoking, alcohol intake,chronic conditions, underweight, previous treatment of TB, adverse drug reaction, and treatment outcome and the development of DR-TB. Odds ratios were computed to determine the strength of the association between the dependent (DR-TB) and the independent variables. Logistic regression was used to investigate the overall effect of the independent variables on the outcome. Additionally, a multinomial logistic regression was used to determine the association between underweight DR-TB cases and the risk of developing adverse drug reactions and treatment outcomes. These uncovered the true effects of each independent variable on the outcomes. The results were presented in the form of tables, frequencies, odds ratios, and relative risk ratios (RRR) with a 95% confidence interval at a P-value<5%.

### Results

This section presents the results of the study conducted among DR-TB cases recorded from 2018 to 2022 in four regions of Ghana. In the study, a total of 3,430 cases were recorded in the study.

According to Table 1's findings, more cases (27.4%) are between the ages of 30 and 39. 17.6% of the cases involved people under 20 years old, and 23.9% involved people in their 40s and 49s. The report also reveals that men make up 69.5% of the cases, and that 58.3% of them are single. In terms of education, those without a formal education make up 31.3%, those with a Junior High School (JHS) certificate make up 33.9%, and only 10.0% make it up to the tertiary level.

The results show that the total number of normal TB cases is 86.2%. The overall prevalence of DR-TB cases is 13.8%, out of which 6.7% of them are underweight. Out of this, 10.9% are smokers, 37.1% drinkers, 8.4% formed previously treated TB cases, and 30.9% had chronic conditions. The percentage of those who experienced minor adverse drug reaction was 17.9%, those with major adverse drug reaction were 1.8%, and the majority (80.2%) did not record any adverse drug reaction. Regarding treatment outcomes, many of the reported cases (34.6%) were cured, those who completed their treatment (48.5%), defaulters (1.2%), and those who died (4.2%). The prevalence of underweight DR-TB in the four study sites is 6.7%.

According to the findings in the Ashanti region, the prevalence of DR-TB increased gradually between 2018 and 2022, as shown in Fig 1. The prevalence of DR-TB in the Ashanti region was 8.2% in 2018, and it steeply declined from 2019 of 4.5% to 10.1% in 2022. The prevalence in the Eastern region was 10.6% in 2018 and rose to 19.8% in 2019. But it dropped to 12.2% in 2020, 8.8% in 2021, and 5.3% in 2022. Again, the Central region recorded the highest rate of 42.1% in 2018, increased sharply to 55.4% in 2019 and declined continuously to 27.8% in 2022. A similar decline is observed in Upper West Region (UWR) which started with 8.9% in 2018, increased to 22.4% in 2019 and declined to 2.7% in 2022.

**Table 1. Socio-demographic and clinical risk factors of the study population.**

| Age | Freq. | Percent | Category of TB | Freq. | Percent |
|---|---|---|---|---|---|
| <20 | 605 | 17.6 | Normal | 2,957 | 86.2 |
| 20-29 | 304 | 8.9 | DR-TB | 473 | 13.8 |
| 30-39 | 940 | 27.4 | **Previously treated** | | |
| 40-49 | 822 | 23.9 | No | 3,142 | 91.6 |
| 50-59 | 439 | 12.8 | Yes | 288 | 8.4 |
| >or=60 | 320 | 9.3 | **Smoking** | | |
| **Sex** | | | No | 3,056 | 89.1 |
| Male | 2,385 | 69.5 | Yes | 374 | 10.9 |
| Female | 1,045 | 30.5 | **Alcohol intake** | | |
| **BMI** | | | No | 2,157 | 62.9 |
| 18.5 to 60 | 1,932 | 56.3 | Yes | 1,273 | 37.1 |
| <18.5 | 1,498 | 43.7 | **Chronic condition** | | |
| **Marital status** | | | No | 2,371 | 69.1 |
| No | 1,998 | 58.3 | Yes | 1,059 | 30.9 |
| Yes | 1,432 | 41.8 | **Adverse drug reaction** | | |
| **Education level** | | | Nil | 2,750 | 80.2 |
| Tertiary | 343 | 10.0 | Major | 63 | 1.8 |
| SHS | 419 | 12.2 | Minor | 617 | 17.9 |
| JHS | 1,162 | 33.9 | **Treatment outcome** | | |
| Primary | 434 | 12.7 | Cured | 1,188 | 34.6 |
| Nil | 1,072 | 31.3 | Completed | 1,665 | 48.5 |
| **Underweight** | | | Died | 143 | 4.2 |
| Underweight DR-TB | 231 | 6.7 | Relapsed | 2 | 0.1 |
| Not underweight DR-TB | 242 | 7.1 | Defaulted | 40 | 1.2 |
| | | | Referred | 1 | 0.0 |
| | | | LTFU | 376 | 10.9 |
| | | | Failure | 15 | 0.4 |

The results shown on Table 2 presents the regional representation of the prevalence of DR-TB cases in the study areas in Ghana. The Central Region recorded the highest prevalence of 42.6% and Upper West Region recorded the lowest of 7.9%. However, the overall prevalence rate was 13.8%.

In Table 3, it is realized that age is generally statistically significant and associated with the development of DR-TB. The study shows that all the age categories, 20–29, 30–39, 40–49, 50–59, and>or=60 years, are 4.40, 1.94, 3.24, 4.29, and 5.77 times increased odds of DR-TB, respectively, compared to those below 20 years. After adjusting for smoking and alcohol intake, the odds of developing DR-TB according to the age ranges listed increased to 5.01, 2.76, 4.84, 6.04, and 6.68, respectively, compared to those below 20 years.

Besides, BMI was statistically significant and associated with DR-TB. Cases with BMI < 18.5 were 1.27 times increased odds of DR-TB compared to those with normal BMI. However, it became statistically insignificant after controlling for smoking and alcohol intake.

Additionally, TB cases that were married and those that drink alcohol were statistically significant respectively and associated with DR-TB. It shows that the married class has 1.88 times increased odds of DR-TB compared to the single and those who drink alcohol were 0.39 times the odds of DR-TB compared to those who do not. After adjusting for smoking and alcohol intake, the married class still had increased odds (OR 3.58) of DR-TB compared to the unmarried

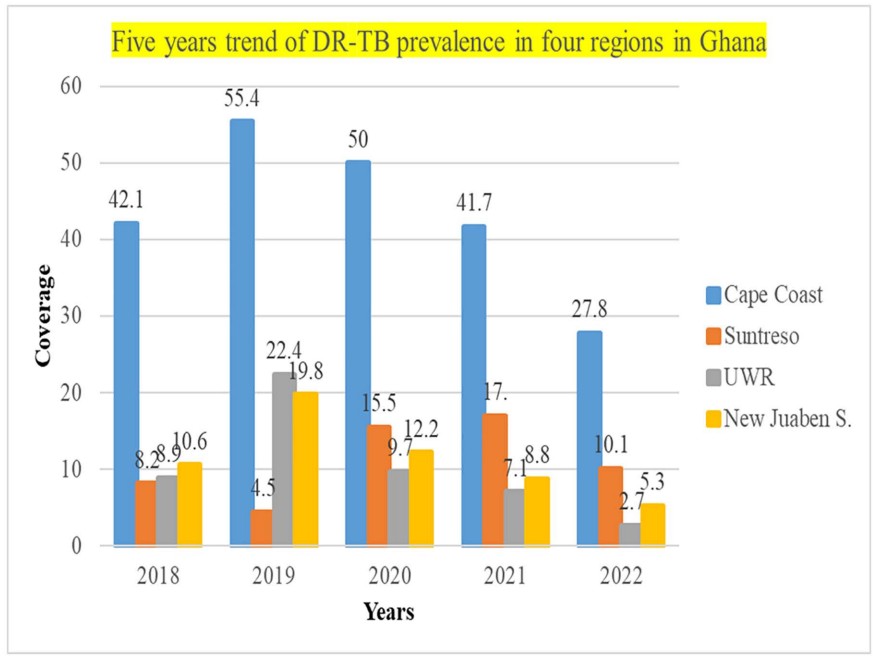

**Fig 1. Prevalence of DR-TB in four regions of Ghana – Five years trend.**

**Table 2. Prevalence of DR-TB in the four regions of Ghana.**

| | | Ashanti (Suntreso) | Eastern (New Juaben South) | Upper West | Central | Total |
|---|---|---|---|---|---|---|
| Category of TB | Normal TB | 1386 | 721 | 679 | 171 | 2957 |
| | DR-TB | 189 (12%) | 99 (12.1%) | 58 (7.9%) | 127 (42.6%) | **473 (13.8%)** |
| **Total** | | **1,575** | **820** | **737** | **298** | **3,430** |

group. Additionally, we controlled for previously treated for TB only and it shows an association with 0.49 times the odds of DR-TB compared to those who were not.

In the arena of education, the results show that cases with SHS, primary education, and not formally educated were statistically significant and associated with DR-TB. Cases with SHS, Primary, and Nil were 2.74, 1.98, and 2.53 times increased odds of DR-TB compared to those who attained tertiary level of education. After controlling for smoking and alcohol intake, cases with SHS level of education were still statistically significant with an increased odds of 2.09 times compared to those who had tertiary education.

Among the previously treated TB cases, the results show a statistically significant association with DR-TB. Those who were previously treated for TB have 24.38 times increased odds of DR-TB (P-value<0.00) compared to those who were not. Also, after adjusting for smoking and alcohol intake, the odds of developing DR-TB increased by 22 folds (22.03) compared to those who were not previously treated for TB.

The results also show that adverse drug reaction as a factor was statistically significant and associated with the development of DR-TB. It means participants that developed major and minor adverse reactions were 156.11 (P-value<0.00) and 21.77 (P-value<0.00) times, respectively, increased odds of developing DR-TB compared to those without adverse

**Table 3. Univariable and multivariable logistic regression analysis of the socio-demographic and clinical risks factors influencing DR-TB.**

| DR-TB | COR | P-value | [95% CI] | aOR | P-value | [95% CI] |
|---|---|---|---|---|---|---|
| **Age** | Reference=<20years | | | | | |
| 20-29 | 4.40 | <0.00 | 2.79-6.94 | 5.01 | <0.00 | 3.17–7.92 |
| 30-39 | 1.94 | 0.01 | 1.28-2.95 | 2.76 | <0.00 | 1.80–4.22 |
| 40-49 | 3.24 | <0.00 | 2.17-4.85 | 4.84 | <0.00 | 3.21–7.31 |
| 50-59 | 4.29 | <0.00 | 2.80-6.59 | 6.04 | <0.00 | 3.91–9.34 |
| >or=60 | 5.77 | <0.00 | 3.72-8.94 | 6.68 | <0.00 | 4.29–10.39 |
| **BMI** | Reference=18.5 to 60 | | | | | |
| <18.5 | 1.27 | 0.02 | 1.05-1.55 | 1.17 | 0.122 | 0.96–1.43 |
| **Marital status** | Reference=Single | | | | | |
| Married | 1.88 | <0.00 | 1.54-2.28 | 3.58 | <0.00 | 2.86–4.48 |
| **Alcohol intake** | Reference=No | | | | | |
| Yes | 0.39 | <0.00 | 0.31-0.49 | 0.49 | <0.00 | 0.38–0.63 |
| **Educational level** | Reference=Tertiary | | | | | |
| SHS | 2.74 | <0.00 | 1.74-4.32 | 2.09 | 0.01 | 1.21–3.63 |
| JHS | 1.07 | 0.76 | 0.69-1.66 | 0.36 | <0.00 | 0.22–0.59 |
| Primary | 1.98 | 0.01 | 1.24-3.16 | 0.59 | 0.05 | 0.35–0.99 |
| Nil | 2.53 | <0.00 | 1.67-3.84 | 1.10 | 0.68 | 0.69–1.75 |
| **Previously treated** | Reference=No | | | | | |
| Yes | 24.38 | <0.00 | 18.41-32.27 | 22.025 | <0.00 | 16.58–29.26 |
| **Adverse drug reaction** | Reference=No | | | | | |
| Major | 156.11 | <0.00 | 72.67-335.34 | 125.50 | <0.00 | 58.05–271.34 |
| Minor | 21.77 | <0.00 | 17.06-27.78 | 23.59 | <0.00 | 18.32–30.39 |
| **Treatment outcome** | Reference=Referred | | | | | |
| Cured | 0.57 | <0.00 | 0.45- 0.72 | .52 | <0.00 | 0.41–0.66 |
| Completed | 11.46 | <0.00 | 7.82 −16.79 | 9.67 | <0.00 | 6.56–14.28 |
| Died | 5.79 | 0.22 | 0.36 −92.98 | 4.51 | 0.29 | .28 −72.43 |
| Relapsed | 3.12 | 0.01 | 1.59- 6.09 | 2.62 | 0.01 | 1.33–5.18 |
| LTFU | 0.39 | <0.00 | 0.25 - 0.62 | .45 | <0.00 | 0.29–0.70 |
| Failure | 37.63 | <0.00 | 8.42 −168.17 | 35.24 | <0.00 | 7.76–159.99 |
| **Chronic condition** | Reference=No | | | | | |
| Yes | 0.398 | <0.00 | 0.31 - 0.51 | 0.47 | <0.00 | 0.36–0.62 |

**Adjusted for: Smoking and alcohol intake; and previously treated for TB only. COR: Crude odds ratio, aOR: Adjusted odds ratio**

drug reactions. After adjusting for smoking and alcohol intake, the odds of developing DR-TB as a result of major and minor adverse reactions still increased by 125.50 (P-value<0.00) and 23.59 (P-value<0.00) times, respectively, compared to those without adverse drug reactions.

Also, for treatment outcome, those who got cured (COR 0.57, P-value<0.00, 95% CI 0.45–0.72), those who completed treatment (COR 11.46, P-value<0.00, 95% CI 7.82–16.79), relapsed cases (COR 3.12, P-value=0.01, CI 1.59–6.09), lost-to-Follow-up (LTFU) cases (COR 0.39, P-value=0.01, CI 0.25–0.62), and those who had treatment failure (COR 37.63, P-value<0.00, 95% CI 8.42–168.17) were all statistically significant and associated with DR-TB compared to those who were referred. After adjusting for smoking and alcohol intake, those who got cure (aOR 0.52, P-value<0.00, CI 0.41–0.66), those who completed treatment (aOR 9.67, P-value<0.00, CI 6.56–14.28), those who were LTFU (aOR 0.45,

P-value<0.00, CI 0.29–0.70), and those who had treatment failure (aOR 35.24, P-value<0.00, CI 7.76–159.99) were all statistically significant and associated with DR-TB.

The multinomial logistic regression in Table 4 explains the association between underweight DR-TB cases and the occurrence of adverse drug reactions and treatment outcomes. The associations are all statistically significant with P-values <0.00. The results show that underweight DR-TB cases are 5.74 times more likely to experience adverse drug reactions compared to the normal TB cases. Also, underweight DR-TB cases are 21% less likely to have a favourable treatment outcome compared to the normal TB cases. Similarly, the DR-TB cases that were not underweight equally showed an association between adverse drug reactions and the treatment outcome, and the associations were statistically significant (P-value<0.00). The results show that, though these cases were not underweight, they had a relative risk ratio (RRR) of 5.04 times more likely to develop adverse drug reactions and 19% less likely to have a favourable treatment outcome compared to the normal TB cases.

## Discussion

In this study, the majority of the TB cases (69.5%) were males, which is consistent with other studies [15–18]. This is also in conformity with the worldwide epidemiology of male dominance in TB cases [17].This probably shows that the males are mostly engaged in risky lifestyles and hence are more prone to developing TB and DR-TB as compared to the females. This does not necessarily mean that by being a male increases their level of risks but the exposure determines the level of risks. Additionally, this study shows that the majority of the DR-TB cases were aged over 20 years, which agrees with another study conducted in Ghana [18] and Pakistan [19]. Generally, the older the age, the weaker the immune system and the chance of developing comorbidities increase which may lead to an increase in contracting DR-TB [20]. The study shows that 34.6% of the reported TB clients got cured, 48.5% of the cases completed treatment, 10.9% LTFU, and 4.2% of the cases died. This is in conformity with a study conducted in Pakistan that also show similar findings [19]. The high LTFU rate of 10.9% coupled with 4.2% that died might have contributed to the low cure and treatment completion rates in the four regions [19].

The findings in this study show that Cape coast in the Central region recorded the highest prevalence of DR-TB in 2019 with 55.4%, and the lowest was in 2022 with 27.8%. The Ashanti, represented by Suntreso recorded the highest prevalence of 15.5% in 2020 and the lowest of 8.2% in 2018. The prevalence rate has decreased in all the study sites from 2018 to 2022. However, conscious efforts must be put in place to properly manage the identified cases and reduce the case load further. Also, in the Eastern region, the highest prevalence was recorded in 2019 at 19.8% and the lowest at 5.3% in 2022. From the findings, the cases in the Eastern region are declining. It may mean that the cases are being managed well or that more cases are not being detected [21]. This is because some of the GeneXpert machines in the Eastern region have broken down, and therefore, adequate testing could not be done for suspected DR-TB cases. Additionally, the Upper West Region had the least number of cases, with a prevalence rate of 2.7% in 2022 and the highest rate of 22.4% in 2019, which is still lower in comparison with other studies [22–24].

Table 4. Multinomial logistic regression on association between underweight DR-TB cases and adverse drug reaction and treatment outcome.

| Underweight DR-TB | RRR | P-value | [95% CI] |
|---|---|---|---|
| Normal TB | Base outcome | | |
| **Underweight DR-TB** | | | |
| Adverse drug reactions | 5.74 | <0.00 | 4.86–6.79 |
| Treatment outcome | 0.79 | <0.00 | 0.74–0.86 |
| **Not underweight DR-TB** | | | |
| Adverse drug reactions | 5.04 | <0.00 | 4.29–5.90 |
| Treatment outcome | 0.81 | <0.00 | 0.75–0.87 |

The COVID-19 pandemic may have had an impact on the DR-TB case detection in this study for the 2020–2022 period. Numerous studies have demonstrated that TB and DR-TB cases, which typically have symptoms with COVID-19, are less likely to seek medical attention in hospitals as a result of the COVID-19 pandemic and the stigma associated with it [23–28]. Furthermore, the COVID-19 pandemic may have reduced TB funding and resource mobilization, which likely contributed to low case detection [26]. The consequence is that, all these factors may lead to an increase in DR-TB prevalence after the pandemic [27]. That could be the reason for the steady drop in DR-TB prevalence during the COVID-19 era [28].

The overall prevalence of DR-TB in 2022 was 13.8%. DR-TB case detection rate maybe d low due to inadequate GeneXpert machines to detect the cases. A systematic review and meta analysis revealed a pooled prevalence rate of 11.6% [22] which is lower than what is reported in this study. Also, the prevalence of MDR-TB among newly diagnosed TB cases in another systematic review and meta analysis was 4% [29] and Sudan reported the highest prevalence of 20% [23]. The difference in prevalence rate among the different studies may arise due to differences in lifestyle risks. A study conducted in Eastern Sudan shows a DR-TB prevalence rate of 39% [30] and a global estimate of 2.9% among new cases [31]. The Eastern Sudan prevalence rate is higher than the prevalence recorded in this study probably because in Sudan for instance, about 20–40% of the population have no access to health services coupled with insecurity in the conflict zones [32]. Besides, according to study findings, the National prevalence of MDR-TB in Sudan shows an increasing trend [33]. In another study, DR-TB prevalence rate in Ghana was reported to be 25.2% [34] which is higher than what is reported in this study.

The findings in this study show that age, marital status, SHS level of education, alcohol intake, previously treated cases, major adverse drug reaction, minor adverse drug reaction, and underweight are all factors associated with the development of DR-TB. Also, treatment outcome such as; cure, completed treatment, relapsed, LTFU, and treatment failure are also associated with DR-TB. For age, the findings show that all the age categories, 20–29, 30–39, 40–49, 50–59, and>or=60 years increase the odds of developing DR-TB by about 5.01, 2.76, 4.84, 6.04, and 6.68 times respectively compared to those below 20years even after adjusting for smoking and alcohol intake. This is in consonance with the findings in Saudi Arabia [35] and Brazil [36]. However, the extent of association is higer in this study compared to that conducted in Saudi Arabia [35]. This maybe due to the economic conditions that prevail in Saudi Arabia which might have led to improved immune system compared to that in Ghana. Also, this study involved four regional DR-TB treatment centers with a higher coverage than that of the study in Saudi Arabia which was conducted in only one hospital with a sample size of 154 compared to 3430 in this study conducted in Ghana.

Additionally, this study shows that alcohol intake is associated with lower odds of (0.49) DR-TB compared to those who do not take alcohol even after adjusting for previously treated TB cases. This is contrary to this study [35] which shows higher odds (72.1) of DR-TB. The variation in extent of association maybe related to the sample size between the two study areas which may influence the results. The study in Saudi Arabia used a smaller sample size compared to this study. Additionally, this study could not measure the specific amounts of alcohol consumed per person because of the use of secondary data.

Education at the SHS level is associated with DR-TB. In the adjusted logistic regression, TB cases that completed SHS level of education are 2.09 times increased odds of DR-TB compared to those without formal education. This agrees with the findings conducted in Brazzil [37]. The SHS level students are mostly adolescents and young adults with youthful exuberance. Therefore, they are most likely to be more involved in risky behaviours and that may expose them to getting infections including DR-TB [38].

Moreover, this study shows that marital status increases the odds of DR-TB by 3.58 times which is consistent with a study conducted in Pakistan [19] which reveals that marital status increases the odds of DR-TB by 2 folds. This could be due to the fact that if anyone partner gets affected, the other partner is likely to contract the disease from the other partner, thus increasing the chance of both developing DR-TB.

In this study, the findings show that chronic condition is associated with DR-TB. This is consistent with the study conducted in Saudi Arabia [35] where study participants that had chronic conditions had lower odds (0.47) of DR-TB which is similar to that recorded in Saudi Arabia. This is contrary to other studies that suggest that having chronic condition increases the odds of DR-TB [39] largely due to compromised immunity which makes the individuals affected susceptible to infections. Also, the use of immunosuppressant medications may suppress their immune system further increasing the vulverability to DR-TB.

The findings in this study show that there is an association between underweight and adverse drug reactions (Major and minor drug reactions). In this study, underweight DR-TB cases are 5.74 times more likely to experience adverse drug reactions compared to the normal TB cases. This is in agreement with the finding that shows that underweight TB cases stand a risk of isoniazid resistance, probably due to their inability to tolerate the drugs [40]. This relates well to another study that reveals that DR-TB cases that were underweight had a longer treatment and culture conversion time [41] compared to the normal TB cases. Hence, underweight affects negatively the treatment outcome of DR-TB cases.

Additionally, there is an association between underweight and treatment outcome, which is consistent with a study conducted in South India [42]. Generally, good nutritional status improves health outcomes, medication absorption, and efficacy [43]. Therefore, naturally, underweight reduces the bodies immune function and further exposes the case to all forms of infections [44].

Conversely, a study conducted in Saudi Arabia shows no association between DR-TB and underweight (BMI) [35]. This is probably due to the fact that Saudi Arabia has better economic conditions with a decline in underweight cases and improved health conditions which might have contributed to the findings shown in that study [45]. This study shows that, underweight DR-TB cases are 20.5% times less likely to have a favourable treatment outcome compared to the normal TB cases. This agrees with the findings of a study conducted in Uganda [15] and in Ethiopia [46].

Moreover, in this study, the findings show that previously treated TB cases are 22 times increased odds of DR-TB compared to those who were not previously treated after adjusting for smoking and alcohol intake. This is in consonance with a previous study conducted in Ghana [34]. This is also in conformity with a study conducted in China that shows that previously treated TB cases have higher odds of developing DR-TB [33,41,47]. Additionally, this study agrees with the findings in a study that shows that ineffective treatment of TB is associated with DR-TB [48].

## Limitations of the study

The limitation of this study is based on the source of the extracted data. Part of the data (TB cases) were extracted from the DHIMS2 platform and therefore may record some data entry errors. However, systems are in place to usually edit and validate the data before it is entered into the platform or validated on the platform before it is sent to the server. Again, it was only the available and accessible TB and DR-TB data that were extracted for this study. It is therefore possible that some data could not be retrieved and therefore was not part of this study. The authors could not measure the amount of alcohol comsumed by the DR-TB clients in order to appropriately determine the level of risk associated with it.

## Conclusion

The study shows that the prevalence of DR-TB in Ghana is not too high, not necessarily due to a decline in cases among the population but probably as a result of the country`s inability to detect enough cases for prompt treatment. The study shows that age, marital status, SHS level of education, alcohol intake, previously treated cases, major adverse drug reaction, minor adverse drug reaction, and underweight are all factors associated with DR-TB. Also, treatment outcome such as; cure, completed treatment, relapsed, LTFU, and treatment failure are also associated with DR-TB. Therefore, interventions aimed at improving the nutritional status of DR-TB cases and minimising adverse drug reactions will improve treatment outcomes. Also, promoting healthy lifestyles among people, especially TB cases, will help improve treatment outcomes.

## Supporting information

**S1 File. Available data and materials can be obtained from the corresponding author if required.**
(XLSX)

## Acknowledgments

This is to acknowledge the contribution of Dr. Eric Agboli, and the Medical Directors of the Eastern, Central, Ashanti and Wa Regional Hospitals for their support in the data collection process. We thank the medical superintendents of the participating District hospitals also for their support and cooperation in this study.

## Author contributions

**Conceptualization:** Esther Ba-Iredire, James Atampiiga Avoka, Luke Abanga, Abigail Awaitey Darkie, Emmanuel Junior Attombo, Eric Agboli.

**Data curation:** Esther Ba-Iredire, Luke Abanga, Abigail Awaitey Darkie, Emmanuel Junior Attombo.

**Formal analysis:** James Atampiiga Avoka.

**Funding acquisition:** Esther Ba-Iredire, Abigail Awaitey Darkie, Emmanuel Junior Attombo.

**Investigation:** Esther Ba-Iredire, James Atampiiga Avoka, Luke Abanga, Abigail Awaitey Darkie, Emmanuel Junior Attombo, Eric Agboli.

**Methodology:** Esther Ba-Iredire, James Atampiiga Avoka, Luke Abanga, Abigail Awaitey Darkie, Emmanuel Junior Attombo, Eric Agboli.

**Project administration:** Esther Ba-Iredire.

**Resources:** Esther Ba-Iredire, Luke Abanga, Abigail Awaitey Darkie, Emmanuel Junior Attombo.

**Supervision:** James Atampiiga Avoka, Eric Agboli.

**Validation:** Esther Ba-Iredire, James Atampiiga Avoka, Eric Agboli.

**Visualization:** Abigail Awaitey Darkie.

**Writing – original draft:** James Atampiiga Avoka.

**Writing – review & editing:** James Atampiiga Avoka, Eric Agboli.

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
