## [Decision Letter · Decision Letter 0]

3 Jul 2024

Dear Dr. Avoka,

Thank you for submitting your manuscript to PLOS ONE. After careful consideration, we feel that it has merit but does not fully meet PLOS ONE’s publication criteria as it currently stands. Therefore, we invite you to submit a revised version of the manuscript that addresses the points raised during the review process.

**ACADEMIC EDITOR: **

The authors should address all comments raised by both reviewers (Dr. Sandra Sipetic Grujicic and Dr. Kunchok Dorjee).

In addition to comments given by the reviewers, address the following points (Academic editor comments):

**Methods**

the following points should be clearly identified/corrected:

Clearly identify the study population for determination of prevalence clearly state the model of analysis that help to determine cOR, aOR, p-valuesAs stated in the Manuscript - the main outcome variable was the prevalence of DR-TB in the four study sites (what is the other outcome variable (if any)? and What was the possible categories in prevalence determination?)- As stated in the Manuscript - all susceptible TB cases were also extracted but as cases of interest. (what was the denominator to calculate the prevalence?)

**Result**

- Title of table1a and 1b not clear (reported TB cases (1038) vs TB cases (839) should be clearly define). The two table can be merged and should have complete title descriptionTable 2,3 & 4 have the same objective - better to re analyze all independent variables with outcome variable and produce one table containing cOR, aOR and p-value rather than three tables

**Discussion**

remove subtitles under discussion section

**Reference**

the references were not exhaustive e.g. Sambas etal was missed

We look forward to receiving your revised manuscript.

Kind regards,

Moges Desta Ormago, Ph.D.

Academic Editor

PLOS ONE

Journal Requirements:

3. In the online submission form, you indicated that "The data underlying the results presented in this study are available from the corresponding author upon request."

5. Please include a copy of Tables 1,2,3 and 4 which you refer to in your text on page 6,7,8 and 9.

Reviewers' comments:

Reviewer's Responses to Questions

**Comments to the Author**

1. Is the manuscript technically sound, and do the data support the conclusions?

Reviewer #1: Yes

Reviewer #2: Partly

2. Has the statistical analysis been performed appropriately and rigorously?

Reviewer #1: Yes

Reviewer #2: I Don't Know

3. Have the authors made all data underlying the findings in their manuscript fully available?

Reviewer #1: Yes

Reviewer #2: Yes

4. Is the manuscript presented in an intelligible fashion and written in standard English?

Reviewer #1: Yes

Reviewer #2: Yes

Reviewer #1: "Prevalence and Factors Influencing Drug-Resistant Tuberculosis in four Regions of Ghana"

The paper is well written, and the topic is current. It is a retrospective study. The data refer to the period 2018-2022. years. It is proposed to publish with minimal corrections:

The title of the paper is short and clear, in accordance with the objectives of the research.

Summary

1. The second sentence should be excluded. Not needed.

2. It is not explained what the abbreviation BMI, SHS means.

3. Is there a significant correlation between minor adverse drug reaction and major adverse drug reaction variable?

4. When writing the p value, please enter ˂0.0001 in all places in the introduction.

5. There is an association between BMI and adverse reactions, please indicate whether lower or higher BMI value. The clear connection described is not enough. It is seen only through the conclusion.

6. Enter commas between the minimum and maximum values of the confidence interval.

Introduction

Short and clear. It answers the described problem under consideration. At the end, the goal of the research was defined. It is necessary that the goal of the research be identical both in the summary and at the end of the introduction.

Methods

Well described. It is necessary to describe and explain BMI in the "data extraction" part, and not to be described in the "data analysis" part. Transfer data about BMI to "data extraction". Define which are dependent and which are independent variables within regression analysis. There are no clearly defined categories for all the variables included in the research, and for outcomes. Also, not all outcomes are explained, e.g. major and minor adverse reactions…….

Define the method of calculating the prevalence….

Results

It is necessary to express all percentages in the text and tables with numbers with one decimal place. Please correct. Attachment titles are not adequate.

Join table one and b. Correct the title of the table 1 and b. For example - Demographic characteristics, habits and other characteristics of persons with TB in 4 regions of Ghana.

Table 2 and 3 have the same title. You must enter a number and percentage for each variable in Tables 2, 3, and 4.

No data is entered on graph 1 regarding the x and y axes.

Under each table enter explanations for abbreviations, statistical tests

Discussion

According to the research results. It would be very important if you explained the data analysis in the "Methods" section, just as you divided the discussion of the results into three parts.

Please remove the first two sentences from the discussion.

Conclusion

Correct

Literature

Number of references greater than 30. Adequately cited.

Correct everywhere the way of citing multiple references in the work. For example (5-9) or (5,7,11).

Reviewer #2: The two main objectives of this study were to determine the prevalence of DR-TB and factors associated with development of DR-TB in Ghana. The objectives are important. Below are comments I have largely focussed on methods.

Methods:

1) Retrospective analysis of DR-TB data from four regions of Ghana were carried out. To determine prevalence, defining the population base in which the outcome is determined is important - sampling method. Authors need to clearly state how the DR-TB cases were sampled. As written, I had to assume that DR-TB cases were measured among all TB cases. If this is the case, it needs to be stated clearly. presenting at various treatment centers in the four regions.

2) How are the four regions selected. How many regions are there? How many treatment centers were included? Answering these questions are important from generalizability perspective. If a convenience sampling approach was taken, it needs to be stated.

3) The inclusion criteria states that all DR-TB cases in four regions of Ghana. It also states that all susceptible TB cases were also extracted as cases of interest. The phrasing should be revised appropriately. One way to revise may be, 'all DR-TB cases out of all TB cases registered......". If including the susceptible new cases are optional, then how is the prevalence of DR-TB determined? What is the denominator? The exclusion criteria states, 'records of DR-TB cases from 2017 and below and 2023 were excluded'. Improve the english by using the words before 2017 or after 2023 instead of 'below'.

4) If determining risk factors for development of DR-TB is an objective, then including only DR-TB patients cannot allow the determination, as there is no comparator. How the statistical analysis is carried out to determine risk factors is not described. If the authors are meaning to say that sociodemographic and clinical risk factors associated with the development of DR-TB among all TB cases was assessed, then it should be stated clearly that the risk factors were compared between people that did and not have DR-TB.

5) No mention was made of Primary drug resistant TB. Do the DR-TB cases including both Primary and acquired DR-TB cases? If Primary DR-TB is included, then how are the association of adverse drug reactions with development of DR-TB relevant, since these primary DR-TB patients have not received any drug at the time of diagnosis.

6) It appears that diagnosis was happening through Xpert only. Are there any validation by culture or LPA? If diagnosis is only Xpert, RR-TB would be appropriate term.

Results:

1) Overall, the results would need to be better presented.

2) Prevalence of DR-TB was described for 3 regions, Ashanti, Eastern and Central. What is the fourth region?

3) Prevalence of DR-TB should be stratified by Primary Drug resistance or at least DR-TB among new cases and then DR-TB among retreatment cases.

4) What is SHS level of education? Why are more educated people more likely to have DR-TB

Discussion:

I am surprised that the discussion has no mention of the impact of Covid-19 on the case detection of DR-TB.

**Do you want your identity to be public for this peer review?** For information about this choice, including consent withdrawal, please see our Privacy Policy

Reviewer #1: No

Reviewer #2: **Yes: ** Kunchok Dorjee

---

## [Author Response · Author response to Decision Letter 1]

12 Nov 2024

Lower Manya Krobo Municipal Health Directorate

P.O.Box 46

Odumase Krobo-Ghana

31/10/2024

The Editor

PLOS ONE Journal

Dear Sir/Madam,

SUBMISSION OF REVISED MANUSCRIPT TITLED: PREVALENCE AND FACTORS INFLUENCING DRUG-RESISTANT TUBERCULOSIS IN FOUR REGIONS OF GHANA, PONE-D-24-05094

This is to submit to you the responses and revised version of the manuscript number PONE-D-24-05094, titled: Prevalence and factors influencing drug-resistant tuberculosis in four regions of Ghana. One manuscript has the tracked version of the corrections and the other is unmarked.

Kindly take note that the data have been re-analyzed and updated to suit the standards of this journal.

We hope to hear favorably from you soon.

Signed

James Atampiiga Avoka (PhD)

Corresponding author

Methods

• Clearly identify the study population for determination of prevalence

Response

• The study population used for the determination of prevalence was the total number of confirmed TB cases (high-risk group) in the four study sites. Therefore, the prevalence in this case was calculated as all DR-TB cases identified divided by the total number of high-risk populations in the study areas (Confirmed TB cases).

• clearly state the model of analysis that help to determine cOR, aOR, p-values

Response

A logistic regression model was used to determine the crude and adjusted odds ratios and their corresponding p-values at 5% confidence interval.

• As stated in the Manuscript - the main outcome variable was the prevalence of DR-TB in the four study sites (what is the other outcome variable (if any)? and What was the possible categories in prevalence determination?)

Response

The main outcome variable in the study was prevalence of DR-TB. The other objective sought to determine the association between underweight and adverse drug reaction and treatment outcome. Therefore, the other outcome variable was prevalence of underweight DR-TB cases. The prevalence of underweight DR-TB in the four study sites is 6.73%.

- As stated in the Manuscript - all susceptible TB cases were also extracted but as cases of interest. (what was the denominator to calculate the prevalence?)

Response

The denominator was all confirmed TB cases in each study area.

Result

• - Title of table1a and 1b not clear (reported TB cases (1038) vs TB cases (839) should be clearly define). The two table can be merged and should have complete title description

Response

Title on tables 1a and 1b have been made clearer and merged.

• Table 2,3 & 4 have the same objective - better to re analyze all independent variables with outcome variable and produce one table containing cOR, aOR and p-value rather than three tables

Response

All independent variables with outcome variables re-analyzed and combined into one Table.

Discussion

• remove subtitles under discussion section

Response

All subtitles under the discussion section removed.

Reference

• the references were not exhaustive e.g. Sambas etal was missed

Response

References updated to include Sambas et al.

Reviewer 1

Summary

1. The second sentence should be excluded. Not needed.

Response

Second sentence excluded

2. It is not explained what the abbreviation BMI, SHS means.

Response

Abbreviations explained. BMI-Body Mass Index, SHS-Senior High School

3. Is there a significant correlation between minor adverse drug reaction and major adverse drug reaction variable?

Response

There is a significant correlation between DR-TB and major and minor adverse drug reactions. For major adverse drug reaction, (aOR 125.503, p-value <0.000), minor adverse drug reaction (aOR 23.597, p-value<0.000).

4. When writing the p value, please enter ˂0.0001 in all places in the introduction.

Response

P-value corrected.

5. There is an association between BMI and adverse reactions, please indicate whether lower or higher BMI value. The clear connection described is not enough. It is seen only through the conclusion.

Response

For the BMI, we meant underweight-It`s been corrected

6. Enter commas between the minimum and maximum values of the confidence interval.

Response

Corrections effected

Methods

Well described. It is necessary to describe and explain BMI in the "data extraction" part, and not to be described in the "data analysis" part.

Response

BMI explained in the data extraction part.

Transfer data about BMI to "data extraction". Define which are dependent and which are independent variables within regression analysis. There are no clearly defined categories for all the variables included in the research, and for outcomes. Also, not all outcomes are explained, e.g. major and minor adverse reactions…….

Response

All variables explained under study variables.

Define the method of calculating the prevalence….

Response

The prevalence was calculated as all DR-TB cases identified divided by the total number of high-risk populations in the study areas (Confirmed TB cases).

Prevalence= DR-TB cases

Diagnosed TB cases

Results

It is necessary to express all percentages in the text and tables with numbers with one decimal place. Please correct. Attachment titles are not adequate.

Join table one and b. Correct the title of the table 1 and b.

For example - Demographic characteristics, habits and other characteristics of persons with TB in 4 regions of Ghana.

Table 2 and 3 have the same title. You must enter a number and percentage for each variable in Tables 2, 3, and 4.

No data is entered on graph 1 regarding the x and y axes.

Under each table enter explanations for abbreviations, statistical tests

Response

All issues under results have been resolved.

Discussion

According to the research results. It would be very important if you explained the data analysis in the "Methods" section, just as you divided the discussion of the results into three parts.

Response

Data analysis were explained under the data analysis part.

Please remove the first two sentences from the discussion.

Response

First two sentences removed.

Literature

Number of references greater than 30. Adequately cited.

Correct everywhere the way of citing multiple references in the work. For example (5-9) or (5,7,11).

Response

All references corrected.

Reviewer #2:

1) Retrospective analysis of DR-TB data from four regions of Ghana were carried out. To determine prevalence, defining the population base in which the outcome is determined is important - sampling method. Authors need to clearly state how the DR-TB cases were sampled. As written, I had to assume that DR-TB cases were measured among all TB cases. If this is the case, it needs to be stated clearly. presenting at various treatment centers in the four regions.

Response

The issue of prevalence calculation and denominator has been dealt with in the revised version. The study was a retrospective review of DR-TB cases from 2018 to 2022.

2) How are the four regions selected. How many regions are there? How many treatment centers were included? Answering these questions are important from generalizability perspective. If a convenience sampling approach was taken, it needs to be stated.

Response

The study sites were divided into four zones: The Northern, Southern, Coastal, and Middle Belts. The Bono, Western, Ashanti, Brong Ahafo, and Ahafo regions made up the middle belt. The Western and Central Regions were part of the coastal belt. The Northern, Savanna, Upper East, and North East Regions were all part of the Northern Belt. The Eastern, Volta, Oti, and Greater Accra regions made up the southern belt. Following that, a random selection of treatment centers was made from each of the four clusters to include them in the study. Additionally, in each of the selected regions, the central treatment centers were purposefully selected for the study. This is because all the DR-TB cases report to those treatment centers for management.

3) The inclusion criteria states that all DR-TB cases in four regions of Ghana. It also states that all susceptible TB cases were also extracted as cases of interest. The phrasing should be revised appropriately. One way to revise may be, 'all DR-TB cases out of all TB cases registered......". If including the susceptible new cases are optional, then how is the prevalence of DR-TB determined? What is the denominator? The exclusion criteria states, 'records of DR-TB cases from 2017 and below and 2023 were excluded'. Improve the english by using the words before 2017 or after 2023 instead of 'below'.

Response

Corrections effected.

4) If determining risk factors for development of DR-TB is an objective, then including only DR-TB patients cannot allow the determination, as there is no comparator. How the statistical analysis is carried out to determine risk factors is not described. If the authors are meaning to say that sociodemographic and clinical risk factors associated with the development of DR-TB among all TB cases was assessed, then it should be stated clearly that the risk factors were compared between people that did and not have DR-TB.

Response

Adequate description on how the analysis was conducted to determine risk factors has now been added. The study included both DR-TB and normal TB cases.

5) No mention was made of Primary drug resistant TB. Do the DR-TB cases including both Primary and acquired DR-TB cases? If Primary DR-TB is included, then how are the association of adverse drug reactions with development of DR-TB relevant, since these primary DR-TB patients have not received any drug at the time of diagnosis.

Response

This study included only DR-TB cases that were already on treatment. This shows why the study extracted data generated from cohorts 2018 to 2022.Those that were newly diagnosed and not on treatment were not part of this study.

6) It appears that diagnosis was happening through Xpert only. Are there any validation by culture or LPA? If diagnosis is only Xpert, RR-TB would be appropriate term.

Response

Results:

1) Overall, the results would need to be better presented.

Response

The results have been re-analyzed and presented afresh.

2) Prevalence of DR-TB was described for 3 regions, Ashanti, Eastern and Central. What is the fourth region?

Response

The fourth region, Upper West region has now been described.

3) Prevalence of DR-TB should be stratified by Primary Drug resistance or at least DR-TB among new cases and then DR-TB among retreatment cases.

Response

Unfortunately, the data were not disaggregated into new and retreated DR-TB cases. Hence, it would not be possible to determine that at this time.

4) What is SHS level of education? Why are more educated people more likely to have DR-TB

Response

SHS is Senior High School. This group is shown to be more likely to have DR-TB probably due to their youthful exuberance and risky behaviors. Also, the youth are usually carefree and do not adhere to instructions especially during treatment of TB.

Discussion:

I am surprised that the discussion has no mention of the impact of Covid-19 on the case detection of DR-TB.

Response

Discussion on impact of Covid-19 on case detection has now been included.

6) It appears that diagnosis was happening through Xpert only. Are there any validation by culture or LPA? If diagnosis is only Xpert, RR-TB would be appropriate term.

Response

Currently, DR-TB cases in Ghana are detected using the GeneExpert machine as first line and culture and sensitivity test (DST) for the confirmatory test.

---

## [Decision Letter · Decision Letter 1]

2 Jul 2025

Dear Dr. Avoka,

Thank you for submitting your manuscript to PLOS ONE. After careful consideration, we feel that it has merit but does not fully meet PLOS ONE’s publication criteria as it currently stands. Therefore, we invite you to submit a revised version of the manuscript that addresses the points raised during the review process.

We look forward to receiving your revised manuscript.

Kind regards,

Felix Bongomin, MB ChB, MSc, MMed, FECMM

Academic Editor

PLOS ONE

Reviewers' comments:

Reviewer's Responses to Questions

**Comments to the Author**

Reviewer #3: (No Response)

2. Is the manuscript technically sound, and do the data support the conclusions?

Reviewer #3: Partly

3. Has the statistical analysis been performed appropriately and rigorously?

Reviewer #3: Yes

4. Have the authors made all data underlying the findings in their manuscript fully available?

Reviewer #3: Yes

5. Is the manuscript presented in an intelligible fashion and written in standard English?

Reviewer #3: No

Reviewer #3: The study is significant as it highlights the global variation in drug-resistant tuberculosis (DR-TB). It has been conducted on a large scale and represents an important contribution to the field. However, I have some concerns regarding the methodology and the interpretation of the results:

• The manuscript indicates that GeneXpert was the primary method used to validate DR-TB. However, the text does not specify what proportion of the study population underwent this testing. What percentage of the total population was tested for susceptibility to anti-TB drugs?

• Based on the text and the conclusions drawn, I strongly assume that only a small fraction of TB patients underwent these tests. Therefore, the prevalence of DR-TB should be calculated using only those cases that were tested.

• It is inappropriate to discuss alcohol consumption as a risk factor without further clarification, as alcohol consumption can vary widely in terms of quantity. It should be categorized by the amount consumed.

• Marital status or school attendance, considered indicators of close contact status, do not in themselves constitute risk factors for the onset or development of DR-TB. Close contact with an infected individual is a significant risk factor for exposure to the TB bacteria.

• The term "chronic condition" is not well-defined in the manuscript. Chronic diseases can differ greatly and have varying effects on the immune system, either directly or through the medications prescribed for them. They should not be grouped to that further specificity.

• It is unclear what is meant by "male class" and "female class." Is being classified as "male" or "female" itself a risk factor for DR-TB in the absence of other associated risk factors?

**Do you want your identity to be public for this peer review?** For information about this choice, including consent withdrawal, please see our Privacy Policy

Reviewer #3: No

---

## [Author Response · Author response to Decision Letter 2]

4 Jul 2025

Lower Manya Krobo Municipal Health Directorate

P.O.Box 46

Odumase Krobo-Ghana

02/07/2025

The Editor

PLOS ONE Journal

Dear Sir/Madam,

SUBMISSION OF REVISED MANUSCRIPT TITLED: PREVALENCE AND FACTORS INFLUENCING DRUG-RESISTANT TUBERCULOSIS IN FOUR REGIONS OF GHANA, PONE-D-24-05094R1

This is to submit to you the responses and revised version of the manuscript number PONE-D-24-05094R1, titled: Prevalence and factors influencing drug-resistant tuberculosis in four regions of Ghana. One manuscript has the tracked version of the corrections and the other is unmarked.

We hope to hear favorably from you soon.

Signed

James Atampiiga Avoka (PhD)

Corresponding author

PONE-D-24-05094R1

Prevalence and Factors Influencing Drug-Resistant Tuberculosis in four Regions of Ghana

PLOS ONE

• The manuscript indicates that GeneXpert was the primary method used to validate DR-TB. However, the text does not specify what proportion of the study population underwent this testing. What percentage of the total population was tested for susceptibility to anti-TB drugs?

Response

Since the percentage of the population screened for anti-TB medication susceptibility was not one of our variables of interest, it was not calculated. The prevalence of DR-TB and its risk factors were the only things that the writers were interested in.

• Based on the text and the conclusions drawn, I strongly assume that only a small fraction of TB patients underwent these tests. Therefore, the prevalence of DR-TB should be calculated using only those cases that were tested.

Response

The data from the DR-TB treatment centres was used to determine the prevalence of DR-TB. For all presumed TB cases, including DR-TB, the GeneXpert test is currently the primary first-line test in Ghana.

• It is inappropriate to discuss alcohol consumption as a risk factor without further clarification, as alcohol consumption can vary widely in terms of quantity. It should be categorized by the amount consumed.

Response

The authors of this retrospective study only retrieved data that was available at the time of the investigation. The alcohol intake amounts were not included in the records. Most likely, we'll record it as a limitation.

• Marital status or school attendance, considered indicators of close contact status, do not in themselves constitute risk factors for the onset or development of DR-TB. Close contact with an infected individual is a significant risk factor for exposure to the TB bacteria.

Response

Correction: The main risk factors for exposure to tuberculosis are contact with infected partners or classmates, not marital status or attendance at school.

• The term "chronic condition" is not well-defined in the manuscript. Chronic diseases can differ greatly and have varying effects on the immune system, either directly or through the medications prescribed for them. They should not be grouped to that further specificity.

Response

Chronic condition in this study was defined as hypertension, diabetes, HIV and cardiovascular diseases.

• It is unclear what is meant by "male class" and "female class." Is being classified as "male" or "female" itself a risk factor for DR-TB in the absence of other associated risk factors?

Response

One of the main risk factors for DR-TB is being male or female. Males are more vulnerable not only because they are men but also because of their exposure to dangerous behaviours like drinking, smoking, working in hazardous locations, and engaging in poor health-seeking behaviours. In contrast to males, females are more cautious and don't engage in risky behaviours.

---

## [Decision Letter · Decision Letter 2]

19 Aug 2025

Dear Dr. Avoka,

Thank you for submitting your manuscript to PLOS ONE. After careful consideration, we feel that it has merit but does not fully meet PLOS ONE’s publication criteria as it currently stands. Therefore, we invite you to submit a revised version of the manuscript that addresses the points raised during the review process.

**Your revision was evaluated by one the initial reviewers who was not satisfied. I ask you to carefully respond to the major comment of this reviewer and to clarify this issue.**

We look forward to receiving your revised manuscript.

Kind regards,

Igor Mokrousov, Ph.D., D.Sc.

Academic Editor

PLOS ONE

Journal Requirements:

Reviewers' comments:

Reviewer's Responses to Questions

**Comments to the Author**

Reviewer #3: (No Response)

2. Is the manuscript technically sound, and do the data support the conclusions?

Reviewer #3: Partly

3. Has the statistical analysis been performed appropriately and rigorously?

Reviewer #3: Yes

4. Have the authors made all data underlying the findings in their manuscript fully available?

Reviewer #3: No

5. Is the manuscript presented in an intelligible fashion and written in standard English?

Reviewer #3: Yes

**Reviewer #3: My questions have****not**
**been addressed fundamentally.**Most important of them is that the primary objective of the study is to assess the prevalence of drug-resistant tuberculosis (DR-TB) in Ghana, with a conclusion stating that the prevalence is low.

However, the authors did not conduct their own investigation of the prevalence of DR-TB. In their response, they claim that this was not the aim of the study, but this contradicts the text of the manuscript and its conclusion. The authors only refer to prevalence data reported by other sources, which raises concerns about how the data was collected. In a scientific study, it is essential to provide such details clearly.

**Do you want your identity to be public for this peer review?** For information about this choice, including consent withdrawal, please see our Privacy Policy

Reviewer #3: No

---

## [Author Response · Author response to Decision Letter 3]

21 Aug 2025

Lower Manya Krobo Municipal Health Directorate

P.O.Box 46

Odumase Krobo-Ghana

21/08/2025

The Editor

PLOS ONE Journal

Dear Sir/Madam,

SUBMISSION OF REVISED MANUSCRIPT TITLED: PREVALENCE AND FACTORS INFLUENCING DRUG-RESISTANT TUBERCULOSIS IN FOUR REGIONS OF GHANA, PONE-D-24-05094R2

This is to submit to you the responses and revised version of the manuscript number PONE-D-24-05094R2, titled: Prevalence and factors influencing drug-resistant tuberculosis in four regions of Ghana. One manuscript has the tracked version of the corrections and the other is unmarked.

We hope to hear favorably from you soon.

Signed

James Atampiiga Avoka (PhD)

Corresponding author

PONE-D-24-05094R2

Prevalence and Factors Influencing Drug-Resistant Tuberculosis in four Regions of Ghana

Reviewer #3

First question

• The manuscript indicates that GeneXpert was the primary method used to validate DR-TB. However, the text does not specify what proportion of the study population underwent this testing. What percentage of the total population was tested for susceptibility to anti-TB drugs?

Follow-up question

Reviewer #3: My questions have not been addressed fundamentally. Most important of them is that the primary objective of the study is to assess the prevalence of drug-resistant tuberculosis (DR-TB) in Ghana, with a conclusion stating that the prevalence is low.

However, the authors did not conduct their own investigation of the prevalence of DR-TB. In their response, they claim that this was not the aim of the study, but this contradicts the text of the manuscript and its conclusion. The authors only refer to prevalence data reported by other sources, which raises concerns about how the data was collected. In a scientific study, it is essential to provide such details clearly.

Response

Data Sources and Methodology

The study utilized secondary data obtained from the District Health Information Management System 2 (DHIMS2) and existing records from DR-TB treatment registers at selected treatment centers. The prevalence of drug-resistant tuberculosis (DR-TB) was calculated based on TB case data recorded in DHIMS2 and the treatment registers within the study areas.

Importantly, the study did not involve active field screening of clients. Instead, it relied on data already captured through routine health service delivery. DHIMS2 serves as Ghana’s central repository for health data, aggregating monthly indicators from hospitals, health centers, and community-based health planning and services (CHPS) across the country.

TB Screening Policy Context

According to the Ghana Health Service (GHS) policy on TB case detection, all health facilities are expected to screen at least 10% of outpatient department (OPD) attendance for TB. Screening may also be extended to community settings such as churches, mosques, and other public venues when feasible. However, actual screening coverage varies across facilities—some may exceed the 10% threshold, while others may fall short. Due to this variability, the study was unable to determine a reliable percentage of the total population screened in the selected areas. Consequently, this metric was not reported.

Limitations

The prevalence of DR-TB reported in this study reflects only the cases that were tested and documented in the study areas. The study did not conduct susceptibility testing on the broader population potentially exposed to anti-TB drugs.

---

## [Editor Report · Decision Letter 3]

24 Aug 2025

Prevalence and Factors Influencing Drug-Resistant Tuberculosis in four Regions of Ghana

PONE-D-24-05094R3

Dear Dr. Avoka,

We’re pleased to inform you that your manuscript has been judged scientifically suitable for publication and will be formally accepted for publication once it meets all outstanding technical requirements.

Kind regards,

Igor Mokrousov, Ph.D., D.Sc.

Academic Editor

PLOS ONE
---

## [Editor Report · Acceptance letter]

PONE-D-24-05094R3

PLOS ONE

Dear Dr. Avoka,

I'm pleased to inform you that your manuscript has been deemed suitable for publication in PLOS ONE. Congratulations! Your manuscript is now being handed over to our production team.

Kind regards,

on behalf of

Dr Igor Mokrousov

Academic Editor

PLOS ONE